# Development of Agar Substitute Formulated with Mucilage and Pectin from *Opuntia* Local Waste Matter for *Cattleya* sp. Orchids In Vitro Culture Media

Arantza Elena Sánchez-Gutiérrez [1], Genaro Martín Soto-Zarazúa [1,*], Beatriz Liliana España-Sánchez [2], Sarahí Rodríguez-González [3,*] and Sergio Zamora-Castro [4]

1   Faculty of Engineering, Amazcala Campus, The Autonomous University of Querétaro, Carretera, Chichimequillas sin Número Kilómetro. 1, Amazcala, El Marques 76265, Queretaro, Mexico
2   Consejo Nacional De Ciencia y Tecnologia, Center of Technological Research and Development in Electrochemistry, Escobedo 76703, Queretaro, Mexico
3   Faculty of Medicine, The Autonomous University of Querétaro, Queretaro 76120, Queretaro, Mexico
4   Faculty of Engineering, Construction, and Habitat, Universidad Veracruzana, Boca del Rio 94294, Veracruz, Mexico
*   Correspondence: genaro.soto@uaq.mx (G.M.S.-Z.); sarahi.rodriguezg@uaq.mx (S.R.-G.); Tel.: +52-4423329713 (G.M.S.-Z.)

**Abstract:** The technology for reproducing orchids in vitro has had to evolve due to the demand for these plants and the high cost of the biotechnology used due to the agar, the gelling agent. Consequently, research has tended to search for natural substitutes for agar. Our work describes the use of pectin and mucilage hydrocolloids extracted from the local waste matter of two species of *Opuntia (O. ficus-indica* and *O. robusta)* to study as a gelling agent in vitro culture media for *Cattleya* sp. These hydrocolloids were obtained by alkaline hydrolysis. Subsequently, these were used in proportions of 0.8%, 0.6%, 0.4%, and 0.2% in combination with agar to study the gelation time, texture profile analysis (TPA), seed germination under light and dark conditions, and a phenological study, including orchid analyses of leaves and roots, root and leaf length, seedling height, and width of the best-designed treatment were studied. Our results demonstrate that the treatment composed of 0.4% *O. ficus-indica* pectin and 0.4% agar improves the germination time, plant growth, and the number of leaves and roots, resulting in a biostimulant formula for optimal in vitro growth of *Cattleya* sp.

**Keywords:** *Orchid*; local waste-based culture media; nopal hydrocolloids; plant tissue culture media





## 1. Introduction

The orchid is the most popular cut flower in the world. They are appreciated worldwide for their exquisite flowers and are known for their medicinal, food, ritual, and ecological properties [1]. However, the growing demanding market for this plant caused by habitat destruction, modification, and illegal extraction has become a problem. Due to improving the germination of orchid seeds and accelerating the production of orchid plants, agricultural systems have been forced to implement new technologies that improve production [2]. As a result, the in vitro culture technique has emerged. However, in vitro culture is more expensive than traditional plant production methods due to the components used to prepare the culture medium. Commonly, the gelling agent is agar [3]. Agar is a mixture of polysaccharides extracted from the walls of red algae and is the most widely used gelling agent in culture media due to its physicochemical properties, such as thermoreversibility, stability, gel strength, texture, elasticity, and transparency [4]. The gelation of agar occurs only by its agarose content, which is produced exclusively by a hydrogen bond. The agar molecules turned from the random coil into a double helix when the agar solution cooled, then the double helix gathered and generated the hard gel with further cooled. The above coincides with the gelation phenomenon "egg box" produced during

the crosslinking process [5]. However, its use increases the total cost of the medium by 70% due to agar overexploitation and high demand [6]. For this reason, recent studies have focused on research substitutes for agar by exploiting endemic species [7].

*Opuntia* spp., a Mexican endemic species best known as nopal, is a natural polymer that has been a protagonist of different investigations focused on the search for alternatives for the use of waste-based material to expand its market and reduce economic losses due to the organic pads that cannot be sold for human consumption. One of the essential components in *Opuntia* pads is dietary fiber, a rich source of hydrocolloids (pectin and mucilage) [8], well known for their remarkable capacity to capture and retain water. These *Opuntia* hydrocolloids can form gels in combination with the proper solution [9] and have been used to modify mechanical properties in edible films for food applications which allows for improving their properties. Use has also increased [10] for decontamination in water treatment [11], in the construction industry, and as natural coatings [12].

Different substrates, such as polyurethane foam, coconut husk, sawdust, and natural polymers of vegetable origin, like starches and gums, have been investigated due to their high availability in local markets. However, low cost [13] has emerged as the most cost-effective and sustainable alternative that could be used as an alternative for agar in *Cattleya* sp. [14]. An orchid in vitro culture needs a balance of growth regulators as auxins and cytokines in the formations of roots and shoots in plants. Therefore, the Murishage and Skoog (MS) basal media fortified with coconut water and solidified with agar is the most used media for success in developing orchid plants [15,16]. Furthermore, according to Magarelli et al. [17], the *Opuntia* waste, including cladodes and immature fruits, can be potentially employed as biobased culture media for plants. The above contributes to the high availability of resources in Mexico and the low cost of production, suggesting an optimal and highly nutritious raw material for orchid production. It is possible that the innovation aspects associated with the extraction of these compounds (mucilage and pectin) are related to different industries, from biotechnology to construction, since their use modifies the mechanical and chemical properties, providing different characteristics to the products or materials, giving the possibility of being added until obtaining the desired characteristics. In addition to the technological and innovation benefit, its use and recovery are linked to a social benefit since the extraction project is related to nopal producers [18–20]. The focus of our work is related to the optimal exploitation of nopal compounds, with the possibility of performing a guideline to start using them as agar substitutes in national biotechnology, relating this to the increase in national raw materials, which have ecological repercussions and a sustainable economy [21,22].

According to the above, our study describes the optimal extraction of hydrocolloids (pectin and mucilage) from the waste *Opuntia* pads of two species (*O. robusta* and *O. ficus-indica*), formulated as a substitute for the agar, acting as a gelling agent for the optimal in vitro culture media of *Cattleya* sp. growth. For this purpose, alkaline hydrolysis extraction and separation of both hydrocolloids were performed. Also, the agar formulation was obtained under different hydrocolloid concentrations and compared with commercial agar media. Furthermore, the chemical composition of the hydrocolloids, color, gel structure, stability, and mechanical properties was performed. Finally, the agar substitute was validated through the in vitro plant growth of *Cattleya* sp., aiming to develop a functional and low-cost media for the optimal growth of orchid species. The last mentioned by an economic calculation and a comparison with the pectin yield obtained from *Opuntia ficus-indica* is about 1.25 g for 300 g, and the agar yield is 50% for 25 g of dry algae, that is, 12.5 g. Therefore, although we see that the yield of agar is much higher, if you use 0.8 g of agar to gel in each medium, the cost is USD 0.51 with the cost of 454 g for USD 292.44 (CRISOL S.A de C.V. Bacteriological agar), the use of pectin in each culture medium has a cost of USD 0.11 per 0.8 g because 50% agar and 50% pectin are used, we can say that the culture medium has a cost of USD 0.31 per culture medium [23,24].

## 2. Materials and Methods

The pectin and mucilage hydrocolloids were extracted and separated from the *Opuntia* waste cladodes (*O. robusta* and *O. ficus-indica*), which were obtained from agricultural nopal companies, from Ajuchitlan, Colón, Querétaro (*O. robusta*), México and, XATA, Corregidora, Querétaro, México (*O. ficus-indica*).

### 2.1. Extraction and Separation of Hydrocolloids

The extraction of mucilage and pectin was performed according to [24,25]. For this purpose, *Opuntia* cladodes were washed and disinfected with water to remove contaminants before the cladode samples were cut to remove spines and the cuticle. Then, the pieces (300 g) in cubes were placed in distilled water to be heated (1:2 $w/v$) at 85 °C for 20 min to extract the substances that contain the hydrocolloids. Next, the pH of the sample was adjusted to 7–7.5. Subsequently, the piece was milled in a commercial blender and passed through a centrifuge (3500 rpm for 20 min. at 23 °C) to obtain two compounds, the supernatant (mucilage) and the precipitant (pectin).

The precipitant (pectin) was rewashed in NaOH and sodium hexametaphosphate (pH 12) and stirred for 1 h. The solution was filtered through a cotton cloth, and the remaining liquid (pH 2) was centrifuged for 10 min. The filtrate was sedimented for 24 h, refrigerated at 5 °C, and centrifuged to recover the sediment material. The recovered material was distilled (pH 8) with continuous stirring. Ethanol (96%) was added to the solution until the pectin suspension (1:4 $v/v$). The material was recollected and dried in an oven at 60 °C for 16 h. The dried sample was milled until a fine powder was obtained (75 μm).

The supernatant (mucilage) was extracted and put in a solution of NaCl (1 L, 1 M) and stirred for 20 min. Then, ethanol (96%) was added to sediment the mucilage (1:4 $v/v$). The material was recollected and dried in an oven at 60 °C for 16 M. The dried sample was milled until a fine powder was obtained (75 μm).

### 2.2. Agar Gel Preparation

A randomized experimental design was carried out (Supplementary Table S1). The study was performed at 0.8% ($w/v$) of total hydrocolloid concentration, with the proportion of hydrocolloids varying from 0 to 80% ($w/v$) in the mixture. Instead, $Ca^{2+}$ was used at a given concentration to get the desired ionic strength for mucilage. The pH (3.5) range was used for pectin to obtain the gelation. The treatments correspond to combinations of hydrocolloids (mucilage and pectin) tested as solid support alone and combined as partial or total replaced with bacteriological agar (MCDLAB). This study uses agar as a soft gelling agent 0.8% (0.4 g/500 mL). First, the agar media control treatment (CT) was prepared using the procedure described by [20]. Then, aliquots of the liquid medium were added to the individual Petri dish until gelation. In the pectin media, sucrose was mixed with 500 mL of distilled water. Distilled water was added to reach the final volume required, while the pH of this solution was adjusted to 3.5 for pectin gelation conditions. The solution was mixed with the formulation with continuous stirring and heating (75 °C for 20 min). Aliquots of the liquid medium were added to each Petri dish waiting until gelation. The mucilage media and sucrose were mixed in 500 mL of distilled water, and the pH of this solution was adjusted to 5.7–5.8. The solution was mixed with the formulation, and the correct number of ions were added to hot solutions to provide the desired ionic strength $Ca^{2+}$, with continuous stirring, and heating (75 °C for 20 min). Aliquots of the liquid medium were added to each Petri dish waiting until gelation.

### 2.3. Gel Characterization

#### 2.3.1. Chemical Composition of *Opuntia* Hydrocolloids

Fourier transform infrared spectroscopy (FTIR) analysis of two species of hydrocolloids was performed to determine their chemical composition using a Perkin–Elmer Spectrum Gx with attenuated reflectance (ATR) coupled from 4000–400 wavenumber (cm$^{-1}$) and a nominal resolution of 4 cm$^{-1}$.

### 2.3.2. Agar Substitute Stability

Gelation time was qualitatively determined when the mixture ceased flowing; the time and the temperature were measured. Subsequently, the treatments that did not gel were discarded.

### 2.3.3. Agar Substitute Structure and Color

There was the performance of a molecular gelation proposal for the pectin and agar, and mucilage and agar with Marvin software (CHEMaxon, Budapest, Hungary, 2022). When the gels were formed, the color was studied qualitatively, having as a control treatment (CT) the transparent color of the agar.

### 2.3.4. Mechanical Properties of Agar Substitute

Texture profile analyzer (TPA) compression properties of the compound gel systems were also performed with Brookfield's CT3 Texture Analyzer. The samples were placed on the platform of the texture analyzer and compressed by the 36 mm parallel plate at a speed of 1 mm/s to a depth of 5 mm. The uniaxial compression experiments set a load cell of 5 g sensibility. The probe used was spherical. Subsequently, the mechanical parameters measured, including one-cycle hardness, two-cycle hardness, cohesiveness, strength adhesiveness, adhesiveness, springiness, and gumminess, were selected to be culture media.

### 2.3.5. Gelling Agent Validation in *Cattleya* sp. In Vitro

The gel systems with potential characteristics were replicated, adding the requirements for orchid germination to be a media culture. The medium used for seed germination and seedlings growth was Murashige and Skoog (MS), pH 5–5.6, containing organic supplements of 10 mL coconut water, 2 g sucrose, and 0.2 g activated charcoal to trap desired compounds exuded by plants into the gel and promote a slow release of nutrients. A randomized experimental design was carried out with 9 treatments and 6 repetitions. For experiments, 54 seeds were chosen at random and placed two seeds in each of the 27 tubes with culture media in the same way. Once the sowing was finished, three tubes of each treatment were placed in the incubation at room temperature ($25 \pm 1$ °C), with a photoperiod of 16 h of light with an irradiance of 41 $\mu Mm^2 s^{-1}$ and 8 h of darkness and light. The average temperature was 24 °C during the day and 18 °C at night. The other three tubes of each treatment were placed in complete darkness to analyze the difference in the germination process [26–28]. The validation for the asymbiotic seed germination was when it was observed that the embryos emerged from the agar substitute, and forms the protocorm. After this, the number of germinated seeds was quantified 5 and 10 days after being divided in light conditions. After 10 days, the treatments that managed to overcome the phases of germination, elongation of the protocorm, the appearance of leaves, and growth of the stem were taken for the study of the phenological development of the plants, where the variables were evaluated through observations photographing the development of each stage: number of leaves (NL), number of roots (NR), length of root (LR), seedling height (SH), leaf width (LW), leaf length (LL).

### 2.4. Statistics

TPA and compression tests were performed in triplicate, and data were reported using an ANOVA test with the level of the meaningful set at $p$-value $\leq 0.05$. In addition, the data obtained from the variables (NL, NR, LR, SH, LW, and LL) for each of the treatments were evaluated by ANOVA test at $p$-value $\leq 0.05$ with the statistical package SAS version 8.0 (SAS, 1999).

## 3. Results

Since the beginning of the use of biotechnology for in vitro plant reproduction, agar substitutes for stable support in plant reproduction under the most economical laboratory

conditions have been investigated. Investigations on the use of organic and natural products have focused on those that can be used for production and the ease of extraction. In this study, the *Opuntia* pads obtained the following positive results.

The mucilage extracted in the present study was from two- to three-year-old *Opuntia* pads, collected in August—September by water bath and using ethanol (95%) or isopropyl alcohol (95%), with a yield of $1.23 \pm 0.1\%$. For pectin, the yield ranges from 4.42 to 10.39%. However, different authors point out that the pads' mucilage and pectin yield could vary depending on climatic conditions, such as cold and rain, and crop age due to the ability of these polysaccharides to absorb water as a plant's defense mechanism against stress conditions, and the extracting method [29].

### 3.1. Chemical Composition of Hydrocolloids

The FTIR analysis of the mucilage of *O. ficus-indica* and *O. robusta* (Figure 1A) performed in this study shows vibrations in absorption bands located at $3300 \ cm^{-1}$ and $2930 \ cm^{-1}$ for –OH stretching groups, and $1611 \ cm^{-1}$ for the symmetric and asymmetric vibration of COO– bonds, $2975–2919 \ cm^{-1}$ for –CH bond, and $2850 \ cm^{-1}$ for the asymmetric vibration of –CH$_2$ group. However, for *O. robusta*, a slight vibration can be identified at $1730 \ cm^{-1}$ C=O. In addition, they found two bands at 1593 for the functional group COO– and $1388 \ cm^{-1}$ of the COO– bond. They are signals at 1315 and $1246 \ cm^{-1}$. Finally, a band was found at $1030 \ cm^{-1}$ and bands below $1000 \ cm^{-1}$. The FTIR analysis of pectin (Figure 1B) shows a strong band located at $3283 \ cm^{-1}$ for –OH and NH. Subsequently, a band in $2968 \ cm^{-1}$ related with O–CH$_3$ and $1720 \ cm^{-1}$ for functional group C=O. Therefore, there is a region between 1800 for –C(O)–O– bonds and $1500 \ cm^{-1}$ for –CH$_2$ bonds. They followed for the bands between 1250 to $1140 \ cm^{-1}$. The last strong bands were found between 1140 to $1100 \ cm^{-1}$ for the functional group C-O–. The region 800 and $1200 \ cm^{-1}$ is considered the 'fingerprint' region for carbohydrates.

### 3.2. Agar Substitute Stability

The gelling hysteresis agar is defined by the difference between its gelling (38 °C) and melting temperature (85 °C) [27]. In Figure 1C, we can observe the formulated temperature and time relation graphs for gels. It shows gelling a few times when the temperature is closer to boiling. Hence, it has been proven that the gelling temperature is influenced by the methoxylation degree of the C6 of the agarobioses present in the agar and the form of a network [30,31]. Figure 1D presents the colorimetry of the experiment carried out with the formulation designed and cataloged for four types of color: transparent, transparent with residue, transparent opaque, and amber.

The interaction of pectin and mucilage can be explained according to that presented in Figure 2A. It proposed the creation of a covalent bond that will form between the oxygen and the carbon of the agar and mucilage molecules, and the hydrogen is lost as a water molecule, which was caused by the swelling mechanism that occurs at the molecular level, forming linear and flattened reticular structures arranged in such a way that they are in contact, constituting a skeleton or internal structure capable of housing and absorbing water. As shown in Figure 2B, the main structure of the extracted pectin, where homogalacturonan is observed to be a linear chain of 1,4-αD-galacturonic acid (GalA), in which some of the carboxyl groups are methyl esterified, thereby finding esters methyl and esters-o-acetyls. In addition, within its molecules, we find peptide polysaccharides, linear and branched oligosaccharides such as rhamnogalacturonan-I, rhamnogalacturonan-II, α-L-Arabinofuranosyl (araf), and β-D-Galactopyranosil [30]. Furthermore, as shown in Figure 2B it proposed the creation of a covalent bond that will form between the oxygen and the carbon of the agar and pectin molecules. The hydrogen is lost as a water molecule due to the hydrolysis to mix the components.

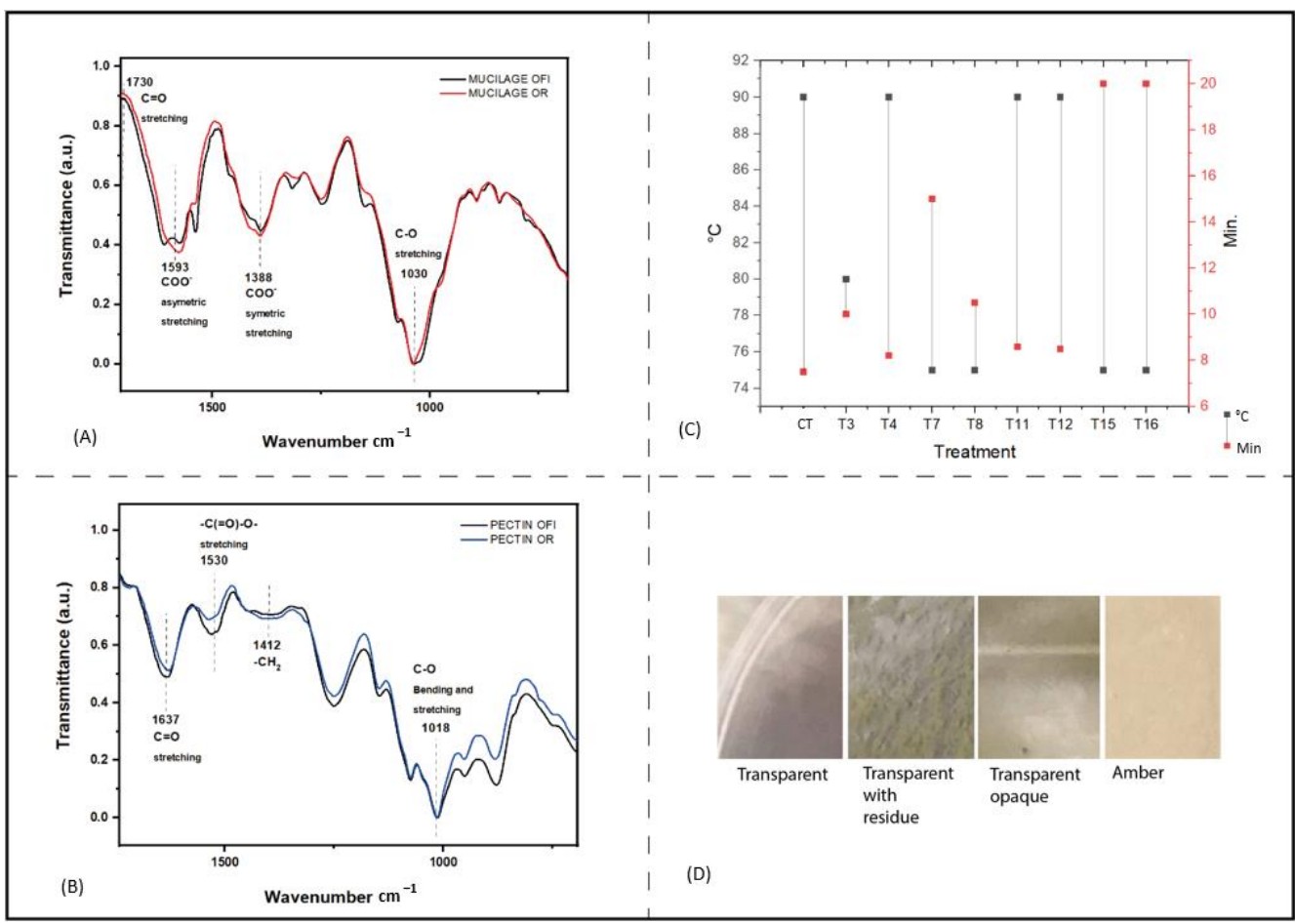

**Figure 1.** Chemical composition of hydrocolloids and agar substitute stability obtained from *O. ficus-indica* and *O. robusta*. (**A**) FTIR spectra of mucilage obtained from *O. ficus-indica* and *O. robusta* (**B**) FTIR spectra of pectin obtained from *O. ficus-indica* and *O. robusta*. (**C**) Temperature and minutes to the gel of designed formulates. CT control treatment. (**D**) Photographs of plates with agar gels with different colors. A catalog of colors was made with experimentation with *Opuntia* hydrocolloids (pectin and mucilage).

### 3.3. Mechanical Properties and Gel Characterization

The treatments discarded did not gel and have in common that the replacement of the agar is total, i.e., mucilage or pectin in a proportion of 0.8%. In the same way, the treatments that did not form a stable support are formulations in which a partial substitution of the agar was made in 6% of the respective hydrocolloid (pectin or mucilage) + 2% of agar. However, the treatments with stable support have in common a partial substitution of agar in proportions of 4% agar + 4% hydrocolloid or greater in proportions of 6% agar + 2% hydrocolloid, regardless of whether the latter is mucilage or pectin. The results suggest that activating the gelling characteristics of pectin and mucilage were possible due to the convenient slow and progressive boiling where it was adequately hydrated, thereby obtaining gelation during the cooling process, activating the C6 and forming hydrogen bonds in combination in proportions of 0.6%, 0.4%, and 0.2% plus agar. Therefore, the treatments are significantly different ($p \leq 0.05$) F value 24.08 [31,32].

Table 1 presents the parameters of different variables studied by TPA analyses, compared with the control treatment.

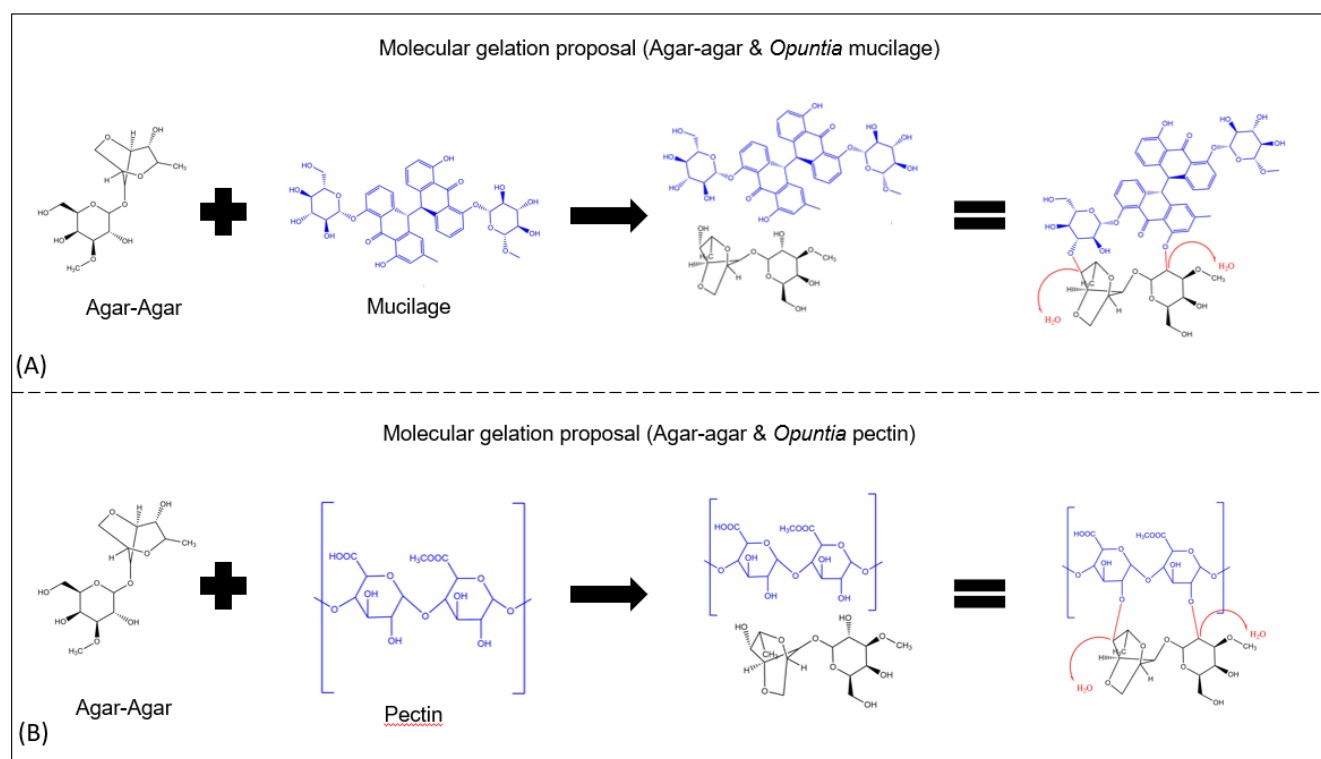

**Figure 2.** Proposal for forming covalent bonds between the molecules of the hydrocolloids extracted from the waste matter of *Opuntia* pads that form a gel. (**A**) Molecular gelation proposal for agar-agar and *Opuntia* mucilage. (**B**) Molecular gelation proposal for agar-agar and *Opuntia* pectin.

**Table 1.** Texture profile analysis (TPA) and mechanical performance of obtained gels.

| Treatments | Hardness | Hardness 2 | Cohesiveness | Adhesive Strength | Adhesive | Gumminess | Elasticity |
|---|---|---|---|---|---|---|---|
| Units | G | G | N | g/cm$^2$ | MJ | G | mm |
| TC | 96.33 ± 0.07 [e] | 30.16 ± 1.41 [e] | 0.35 ± 0.16 [a] | 5.0 ± 0.70 [c] | 0.05 ± 0.03 [a] | 64.40 ± 0.88 [b] | 1.87 ± 0.04 [a] |
| T3 | 72.0 ± 0.86 [f] | 23.0 ± 3.53 [f] | 0.11 ± 0.03 [d] | 5.33 ± 1.06 [c] | 0.09 ± 0.04 [a] | 32.36 ± 1.13 [d] | 1.48 ± 0.21 [c] |
| T4 | 163.33 ± 0.70 [c] | 41.50 ± 1.06 [d] | 0.29 ± 0.16 [a] | 6.33 ± 3.01 [c] | 0.05 ± 0.03 [a] | 52.2 ± 9.26 [b] | 1.52 ± 0.12 [c] |
| T7 | 44.33 ± 1.44 [h] | 15.0 ± 0 [i] | 0.30 ± 0.27 [a] | 3.0 ± 1 [d] | 0.05 ± 0.04 [a] | 19.76 ± 4.45 [e] | 1.03 ± 0.27 [f] |
| T8 | 225.33 ± 2.82 [b] | 99.66 ± 0.70 [a] | 0.33 ± 0.13 [a] | 6.83 ± 0.70 [b] | 0.10 ± 0.06 [a] | 145.53 ± 14.56 [a] | 1.53 ± 0.01 [d] |
| T11 | 54.16 ± 0.35 [g] | 17.83 ± 0.28 [h] | 0.27 ± 0.05 [a] | 4.0 ± 0.5 [cd] | 0.06 ± 0.01 [a] | 37.5 ± 0.92 [c] | 1.59 ± 0.13 [c] |
| T12 | 133.66 ± 3.21 [d] | 71.50 ± 0.5 [c] | 0.28 ± 0.14 [ad] | 5.5 ± 1.32 [c] | 0.06 ± 0.01 [a] | 110.4 ± 35.63 [a] | 1.47 ± 0.14 [c] |
| T15 | 33.83 ± 3.88 [i] | 20.33 ± 1.44 [g] | −0.20 ± 0.55 [c] | 4.5 ± 1.32 [c] | 0.11 ± 0.05 [a] | −18.7 ± 0.49 [b] | 1.20 ± 0.11 [e] |
| T16 | 333.16 ± 2.47 [a] | 60.83 ± 0.70 [b] | 0.18 ± 0.005 [b] | 10.6 ± 1.41 [a] | 0.06 ± 0.01 [a] | 150.03 ± 6.29 [a] | 1.73 ± 0.08 [b] |

Values with different superscripts show significant differences. TC: 100 agar, T3: 50 agar/50 ofip, T4: 80 agar/ 20 ofip, T7: 50 agar/50 ofim, T8: 80 agar/20 ofim, T11: 50 agar/50 orp, T12: 80 agar/20 orp, T15: 50 agar/50 orm, T16: 80 agar/20 orm.

### 3.4. The Functionality of Culture Mediums in Cattleya sp.

The treatments selected for developing in vitro plants were T3, T4, T7, T8, T11, T12, T15, and T16. Treatments selected by their characterization of TPA could be compared with CT. The experiment was carried out through asymbiotic sowing of *Cattleya* sp. seeds in the formulations above (Figure 3A). The experiment was observed 5, 10, and 15 days after. T3 and T11 exhibited germination, rhizoid formation, and protocorm germination in fewer days than CT, regardless of light or dark conditions (Figure 3B). We can observe an increase in the roots of all treatments was observed after 10 days (Figure 3C). However, those treatments fortified with *Opuntia pectin* in a proportion of 50% showed higher growth in roots and leaves compared to CT. We can observe that T3 was the best treatment (Figure 3D) because it was the only treatment that had a significant development through five stages:

no germination, rupture of the test by an enlarged embryo (germination), rhizoid formation, elongation of protocorm (shoot formation), the appearance of the first leaf within shoot region, and last elongation of first leaf [28,33]. Furthermore, it was the only treatment that did not die regardless of weather conditions and generated a more significant number of roots and leaves, in addition to maintaining color. This behavior suggests the shape of the gel network, in addition to water stress and the sugar content in the *Opuntia* pectin.

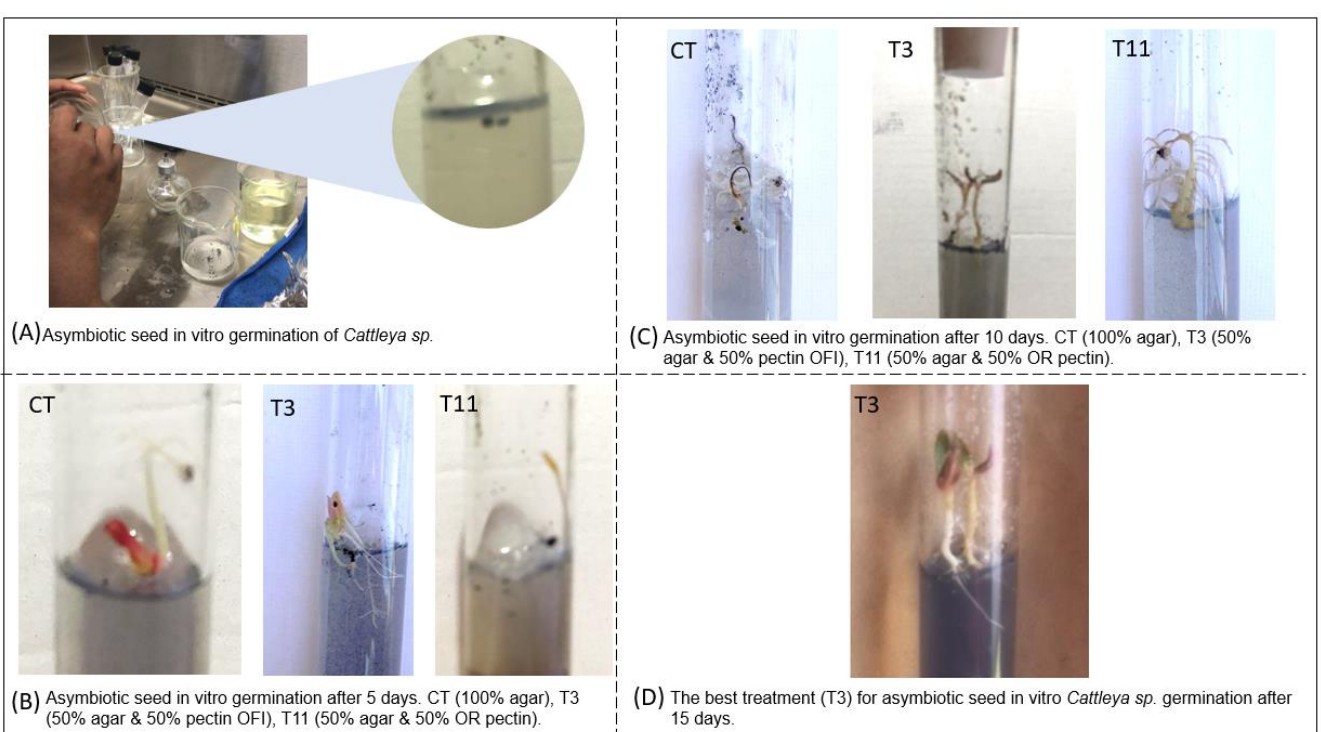

**Figure 3.** Designed Gels of the best treatments. CT (100% agar), T3 (50% agar & 50% pectin OFI), T11 (50% agar & 50% OR pectin). (**A**) Asymbiotic seed in vitro germination of *Cattleya* sp. (**B**) Asymbiotic seed in vitro germination after five days. (**C**) Asymbiotic seed in vitro germination after ten days. (**D**) T3 after fifteen days.

The T3 underwent a phenological study in which the variables evaluated were the number of leaves (NL), number of roots (NR), root length (RL), seedling height (SH), leaf width (LW), and leaf length (LL), after four weeks of sowing in vitro and ANOVA was done with the following results (Table 2).

**Table 2.** The phenological study between the best treatment, after 4 weeks of observation under in vitro growth conditions whit a meaningful level of $p \leq 0.05$.

| Variable | Units | TC (Mean) | Variance | T3 (Mean) | Variance | F Value |
|----------|-------|-----------|----------|-----------|----------|---------|
| NL | Portion | 1.33 | 0.33 | 3.66 | 0.066 | 24.5 |
| NR | Portion | 1 | 0.001 | 4.25 | 0.025 | 14.25 |
| RL | cm | 1.53 | 0.003 | 2.53 | 0.023 | 112.5 |
| SH | cm | 3.73 | 0.41 | 5.97 | 0.13 | 27.48 |
| LW | cm | 0.46 | 0.003 | 0.66 | 0.003 | 18 |
| LL | cm | 1.09 | 0.009 | 1.53 | 0.09 | 5.58 |

(NL), number of leaves, (NR), Root length (RL), Seedling height (SH), Leaf width (LW), and Leaf length (LL).

Table 2 Phenological study for TC and T3.



## 4. Discussion

*Opuntia* waste has been considered a cheap medium for the optimal growth of plants [34]. However, correctly identifying hydrocolloids is the key to determining the optimal concentration of the agar substitute formulation. The FTIR analysis of the mucilage of *O. ficus-indica* and *O. robusta* performed in this study shows that the functional groups present are multiple hydroxyl groups, and the –OH and NH stretching groups involved in the intermolecular bond in carbohydrate and uronic acid molecules as mentioned by [35], with proteins' N–H stretching and bending motions [36]. The –CH groups belonging to the pyranose groups showed more stretching in OFI, than the –CH$_2$ groups of the carboxylic group. The FTIR, showing a lack of waveband at 1749 cm$^{-1}$, is linked to the low degree of esterification, indicating that the carboxyl groups are free and available to interact with water molecules, resulting in their high capacity to absorb water, as well as if the free carboxyl is mixed with Ca$^{2+}$ in the presence of water, they can form a dense structural network, or if they are mixed with others substance, they can form gels, all dependent upon the proportion of these and the preparation conditions.

For *O. robusta*, C=O and COO– bonds of the D-galactopyranosyl uronic acid residues, frequency of aldehydes, and ketones are presented. In addition, symmetric and asymmetric COO– stretching are presented, which confirms the low degree of mucilage esterification attributed to characteristic stretching vibrations of the pyranose ring. Finally, C–O molecules are attributed to the stretching of secondary cyclic alcohols, and the last functional groups present N–H and O–H.

According to the vibrations obtained in the FTIR, we deduce that the mucilage's chemical structure is considered an amorphous substance, characterized by its molecules homogeneously distributed as in the liquid phase. Still, they are immobile. In other words, these do not follow a usually complex structure [37]. The general chemical structure of the mucilage corresponds to heterogeneous polysaccharides with a high content of galactose, mannose, and glucose, mainly acids [38].

Also, the FTIR analysis of the pectin of *O. ficus-indica* and *O. robusta* shows that the functional groups present were –OH and NH stretching groups of alcohol and carboxylic acid involved in inter- and intramolecular hydrogen bonds of the galacturonic acid polymer. The absorption of the O–CH$_3$ extension bonds of the galacturonic acid methyl acetate was due to the stretching of hydroxyl groups. Stretching vibration C=O methyl of the esterified carboxyl groups and vibration related to the stretching of carboxylate ions and the relative ester band, which is more intense in pectin with a high degree of esterification. There were carboxylic acid salts' antisymmetric and COO-symmetric stretching characteristics present and some carboxyl group signals for aromatic ring vibrations [39]. The last region for identifying major chemical groups in polysaccharides was the position and intensity of the bands, which are specific for every polysaccharide. These characteristic bands corresponded to C–O–C stretching, –OH bending, and –CH$_3$ deformation [40]. The chemical structure of pectin is composed of a mixture of highly branched acidic and neutral polymers. Like mucilage, pectin has the property of absorbing a large amount of water.

The color of the agar culture media was transparent, which facilitated the visibility of seed germination. In other aspects, opacity and transparency permit the growth to see bacterial growth in culture [41]. The results were categorized into four colors or variants: transparent, transparent with residues, transparent opaque, and amber. The last, amber, which caused difficulty observing growth and germination, gave a seed. However, the treatments that gelled did not fall into this last category. Further, we can observe that the higher the temperature, the shorter the time for gelation. This was due to the hydrolysis that heat causes in the molecules, thus forming the three-dimensional network that traps the water molecules [42].

The variables studied by the TPA indicated that the hardness resulted in a "semi-hard media" (70 to 200 g/cm$^2$) [43] since when the agar was combined in the same proportions as hydrocolloids, the hardness tended to be ideal for vegetable tissue culture formulations [44]. The cohesiveness resulted in the partial replacement of the agar by decreases in the *Opuntia*

hydrocolloid values. This could be attributed to the fact that the pores of the porous skeletal structures tended to be a more significant and less dense attribute that indicated an internal union of the gel made, allowing with this a better movement of the water molecules which allowed better absorption of this for the germination of the seeds in the tissue culture media. The adhesive force was vital to maintain stable gel support. It is possible to deduce that the adhesive strength depends on the relationship between the amylose and amylopectin contents; the higher the amylose content, the greater the adhesive strength. Therefore, it is inferred that mixing agar and mucilage creates a rigid and robust solid [45]. Adding *Opuntia* hydrocolloids to the agar caused an alteration of the molecules, generating a rupture or fusion depending on the percentage of substitution. Gumminess parameters can range from less than 4 to more than 150 g, depending on and linked to strength and cohesiveness and the material [46]. Agar is classified as a reversible material, which means that it acquires rigidity and, in a few cases, elasticity; the parameters can range from 0 to 2 kPa. It is observed in the results that the hydrocolloids present in the prickly pear losses improve the quality of the culture medium.

The treatments for developing in vitro plants were T3, T4, T7, T8, T11, T12, T15, and T16. Even though the results of the treatments were varied, it was observed that those combined with agar and mucilage form strong gels, which complicated the development and germination of the seeds. Nevertheless, these treatments can be used in biomedical areas for the cultivation of bacteria [47], as substitutes in the food industry to improve the quality of these [48], or even in the cosmetic industry as substitutes for chemical substances [49]. However, those made with agar and pectin turned out to have a stable support that allowed the seeds to absorb water in a better way. Finally, the optimal growth of *Cattleya* sp. that was performed by the T3 formulation (50 agar + 50 OFI pectin) was observed as the best treatment as a stimulator for the growth and generation of plants, linking this to a more remarkable synthesis of sugars through photosynthesis due to the amount of these present in the pectin of *O. ficus-indica* and a large amount of water present that can be absorbed due to its lower hardness [50]. Furthermore, it was observed that the growth of roots in T3 in a biostimulant way forms primary and lateral roots for water and nutrients which were provided in the gel. We can link the biostimulant response to root growth to a large number of nutrients present not only due to the presence of pectin. Furthermore, the length of the roots can be related to the content of flavonoids in pectin that improves the availability of nitric oxide, which directly affects the stem cells responsible for plant root growth [51–54]. The development of the new culture medium (T3) from *Opuntia* and its operating efficiency suggested producing a functional substrate for the optimal growth of orchid species, without modifying the chemical composition or the nutriment requirements for toe orchid growth. The above gives us the guideline to propose research works linked to the growth of not only different genera of orchids, such as *Dendrobium* sp., *Phalaenopsis* sp., or *Oncidium* sp., but also crops that are in danger of extinction, such as the Cactacea genus.

## 5. Conclusions

Hydrocolloids such as mucilage and pectin present in *Opuntia* have been successfully employed as a partial substitute for agar in vitro culture media for stable support. The new gel T3, formulated with 50% agar + 50% OFI pectin, improves the characteristics, showing lesser strength but good stability, less cohesiveness, gumminess, and elasticity. These are guidelines for the best germination, growth, and development of *Cattleya* sp. seeds in vitro, improving germination time and rapid development of roots, leaves, and plant growth.

**Supplementary Materials:** The following supporting information can be downloaded at: https://www.mdpi.com/article/10.3390/pr11030717/s1, Table S1: Gelling Agar agent replaced formulation (partially or fully) for *O. ficus-indica* or *O. rubosta* hydrocolloids.

**Author Contributions:** Conceptualization, A.E.S.-G.; Data curation, A.E.S.-G., B.L.E.-S, S.R.-G. and S.Z.-C.; funding acquisition, A.E.S.-G.; investigation, G.M.S.-Z., B.L.E.-S., S.R.-G. and S.Z.-C.; methodology, A.E.S.-G.; resources, G.M.S.-Z. and S.Z.-C.; supervision, G.M.S.-Z. and S.R.-G.; writing—original draft, A.E.S.-G. and G.M.S.-Z.; writing—review & editing, A.E.S.-G. and B.L.E.-S. All authors have read and agreed to the published version of the manuscript.

**Funding:** The author acknowledges the financial support of CONACYT through the Gran 779027.

**Data Availability Statement:** Not applicable.

**Acknowledgments:** The authors acknowledge Reyna Araceli Mauricio Sánchez (CINVESTAV Unidad Querétaro) for their technical support in FTIR characterization. Acknowledgments to Isabel Nieto, Vanesa Oviedo, and Montse Ramirez (Engineering Faculty, Amazcala Campus. the Autonomous University of Querétaro) for their support in the laboratory experiments.

**Conflicts of Interest:** The authors declare no conflict of interest.

## Abbreviations

| | |
|---|---|
| TPA | Texture Profile Analysis |
| araf | Arabinofuranosyl |
| GalA | Galacturonic Acid |
| Ofip | *Opuntia ficus indica* pectin |
| Ofim | *Opuntia ficus indica mucilage* |
| Orp | *Opuntia robusta pectin* |
| Orm | *Opuntia robusta mucilag* |
| FTIR | Fourier Transform Infrared |

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
