# Peer review of "Development of Agar Substitute Formulated with Mucilage and Pectin from Opuntia Local Waste Matter for Cattleya sp. Orchids In Vitro Culture Media"

_processes, doi:10.3390/pr11030717_

Round 1

Reviewer 1 Report

Please check the attachment (60+ inquiries). Some of those 60 comments are highlighted here, because they are quite significant: 

  1. Cattleya is a genus name, not a species name. Write it as Cattleya sp., and NOT “Orchidea cattleya”. à Check the title of your references, where Cattleya is positioned as a genus.
  2. How about the possibility of using your alternative hydrocolloids for culturing in vitro for other commercial orchid genera, such as Dendrobium sp., Phalaenopsis sp., etc.? Please state your answer in the Discussion section of your manuscript.
  3. Please write “agar” with lowercase a, not with uppercase A.

Locations: check the attachment

4.  Please change comma with dot for the decimal numbers. Location: Check the attachment

5 to 61: Check the attachment

Reviewer 2 Report

Some aspects  was not considered in the methodology, results and disscusion.

They are indicated in the text.

Reviewer 3 Report

Dear authors,

The manuscript entitled “Development of agar substitute formulated with mucilage and pectin from Opuntia local waste matter for Orchidea cattleya in vitro culture media” describes the extraction of two hydrocolloids (pectin and mucilage) from waste of O. robusta and O. ficus-indica. In addition, these were used in proportions of 0,8%, 0,6%, 0,4%, and 0,2% in combination with Agar to be tested as a gelling agent for in vitro culture media at O. cattleya

There are good application and science value. However, there are some significant issues that should be carefully considered.

And it is better if the authors consider the following mentioned remarks and further improve the manuscript before submitting the final version.

 Keywords: Agar Substitutes - these should be changed, because they exist in the title

Introduction: Introduction and objectives are well written but it is necessary to emphasize the innovative aspects of the research.

Lines 82-84: You stated: Finally, the Agar substitute was validated through the in vitro plant growth of Orchidea cattleya, aiming to develop a functional and low-cost media for the optimal growth of Orchid species.

This claim should be supported by an economic calculation, a comparison of the economics of the new gel T3 (formulated with 50% agar + 50% OFI pectin) with solidified agar media

 Materials and Methods:

 2.3.5. Agar substitute validation by the Orchid Cattleya growth

 The manuscript presents only preliminary data of a possible Agar substitute in the initiation phase of in vitro cultures (seed germination under light and dark conditions during 10 days) without presenting other complementary assessments, such as the multiplication phase. Therefore, the whole manuscript should be revised for this aspect.

Results and Discussion

The results and discussion are in general clearly presented.

 Table style: at the top of the table, change the writing from uppercase to lowercase letters (sentece case).

 Line 283 - Table 2 Phenological study for TC and T3 -remove 

Table 2 - specify the unit of measure for variables (root length, seedling height, leaf width, leaf length)

Round 2

Reviewer 1 Report

Review of processes-2186494-v2

This manuscript has been revised significantly and therefore can be accepted.

Note: There are some tiny details to be addressed (can be performed during the proofreading stage), as follows:

Line 281: Please change g/cm2 to be g cm-2

Line 365: Please change g/cm2 to be g cm-2

Line 448: Gracilaria salicornia --> Please write scientific names in italic

Line 498: Gracilaria --> Please write the genus with uppercase G letter.

Reviewer 3 Report

I recommend publishing this paper in its present form, as the authors have covered all the comments made in the previous round.